# VILA-U: A Unified Foundation Model Integrating Visual Understanding and Generation

**Yecheng Wu**[1,2*]   **Zhuoyang Zhang**[2*†]   **Junyu Chen**[1,2]   **Haotian Tang**[2†]
**Dacheng Li**[4†]   **Yunhao Fang**[5†]   **Ligeng Zhu**[3]   **Enze Xie**[3]
**Hongxu Yin**[3]   **Li Yi**[1]   **Song Han**[2,3]   **Yao Lu**[3]
Tsinghua University[1]   MIT[2]   NVIDIA[3]   UC Berkeley[4]   UC San Diego[5]
https://hanlab.mit.edu/projects/vila-u

## ABSTRACT

**VILA-U** is a **U**nified foundation model that integrates **V**ideo, **I**mage, **La**nguage understanding and generation. Traditional visual language models (VLMs) use separate modules for understanding and generating visual content, which can lead to misalignment and increased complexity. In contrast, VILA-U employs a single autoregressive next-token prediction framework for both tasks, eliminating the need for additional components like diffusion models. This approach not only simplifies the model but also achieves near state-of-the-art performance in visual language understanding and generation. The success of VILA-U is attributed to two main factors: the unified vision tower that aligns discrete visual tokens with textual inputs during pretraining, which enhances visual perception, and autoregressive image generation can achieve similar quality as diffusion models with high-quality dataset. This allows VILA-U to perform comparably to more complex models using a fully token-based autoregressive framework. Our code is open sourced at https://github.com/mit-han-lab/vila-u.

## 1 INTRODUCTION

In recent years, large language models (LLMs) have demonstrated superior capabilities in various language tasks. Their appealing properties like instruction following, zero-shot generalization, and few-shot in-context learning motivate researchers to combine them with vision models to build visual language models (VLMs) for multi-modal tasks. Many efforts (Dai et al., 2024; Liu et al., 2024b; Lin et al., 2023) in this field have achieved remarkable performance on visual language understanding. In these works, visual inputs are projected onto LLMs' semantic space through a vision model like CLIP (Radford et al., 2021) to bridge two modalities by including text-image alignment objectives.

In addition to visual understanding, another essential research direction in combining visual and language modalities is visual generation. There are two popular approaches for text-guided image generation. One approach employs diffusion models (Rombach et al., 2022a), a powerful tool for various generation tasks. The other line of work converts visual content into discrete tokens through vector quantization (VQ) and then leveraging autoregressive transformers for high-quality and diverse generation (Esser et al., 2021; Yu et al., 2021; Lee et al., 2022; Tian et al., 2024b; Sun et al., 2024).

Witnessing the rapid advancements in both visual understanding and generation, an emerging trend is to unify these techniques into a single multi-modal framework. Prior to VILA-U, there are two main approaches to achieving such unification: (1) One approach (Liu et al., 2024a; Yu et al., 2023a; Xie et al., 2024) utilizes a VQGAN-based (Esser et al., 2021) tokenizer to convert visual inputs into discrete tokens and leverages an autoregressive model for both understanding and generation. However, (Xie et al., 2024) has shown that visual tokens from VQGAN-based encoder lack semantic information and usually results in a severe performance drop in downstream visual understanding tasks. (2) Another approach (Zhan et al., 2024; Ge et al., 2023b; Jin et al., 2023) utilizes a codebook to quantize features produced by a pre-trained vision model like CLIP. Since CLIP features encode

---

*Equal Contribution.
†Part of the work done during an internship at NVIDIA.

rich semantic information, these approaches generally achieve significantly better performance on understanding tasks. However, these tokenizers lack decoding capability, requiring an external visual generation model, such as a diffusion model, to use the generated visual tokens as conditions for producing visual outputs. This approach adds complexity to infrastructure design. Available large-scale foundation model training pipelines and deployment systems have already been highly optimized for language modeling with next-token prediction. Designing and maintaining an additional stack to support diffusion models would incur significant engineering costs.

In this work, we present **VILA-U**, an *end-to-end autoregressive* framework with a unified next-token prediction objective for both visual and text inputs that can achieve competitive performance on both visual language understanding and generation tasks, without the help of external components like diffusion models. We identify two critical principles to unify vision and language modalities: (1) Existing unified end-to-end autoregressive VLMs cannot achieve competitive visual understanding performance because the discrete VQGAN tokens are trained solely on image reconstruction loss and are not aligned with textual inputs. Therefore, it is crucial to introduce text alignment during VQ vision tower pretraining to enhance perception capabilities. (2) Autoregressive image generation can attain similar quality as diffusion models if trained on high-quality data with sufficient size. Guided by these insights, VILA-U features a unified foundation vision tower that converts visual inputs into discrete tokens through vector quantization and aligns these tokens with textual inputs using contrastive learning. The multi-modal training of VILA-U takes advantage of a unified next-token prediction objective for both visual and textual tokens on a small-size high-quality image-text corpus.

We evaluate VILA-U on common visual language tasks, including image-language understanding, video-language understanding, image generation and video generation. VILA-U significantly narrows the gap in visual understanding performance between end-to-end autoregressive models and continuous-token VLMs, while introducing competitive *native* visual generation capabilities.

## 2 RELATED WORK

**Large Language Models (LLMs).** LLMs based on pre-trained large-scale transformers (Vaswani et al., 2017) has drastically revolutionized natural language processing field. Featuring gigantic model size and pre-training data corpus, LLM has achieved remarkable performance on various linguistic tasks. The development of open-source LLMs such as LLaMA (Touvron et al., 2023a), Mixtral (Jiang et al., 2024) and Vicuna (Chiang et al., 2023) has furthered nourished research on how to adopt LLM for complex language tasks. Besides excellent zero-shot generalizability to diverse domains, LLM is commonly finetuned on custom datasets for better performance on specific tasks. Instruction tuning (OpenAI, 2023; Chung et al., 2024; Ouyang et al., 2022) also stands as a key step for better outputs in applying LLMs. In this work, we adopt the LLaMA-2-7B (Touvron et al., 2023a) model as our basic LLM.

**Visual Language Models (VLMs).** Combining computer vision and natural language processing gives rise to VLM in this LLM era. In VLMs, researchers leverage vision foundation models such as CLIP (Radford et al., 2021), BLIP (Li et al., 2022) and CoCa (Yu et al., 2022) to extract visual features, align with texts, and feed them into LLM to achieve the cross-modality understanding between texts and visual content. Building upon such progress, many VLMs (Alayrac et al., 2022; Li et al., 2023b; Liu et al., 2024b; Lin et al., 2023; Luo et al., 2024; Tian et al., 2024a) have been designed and trained on extensive vision-language data to achieve remarkable performance on visual understanding and reasoning tasks. In this work, we aim to develop a VLM with visual understanding capacities comparable to prior works, while also possessing the new capacity of visual generation.

**Unified Visual Language Models.** Numerous efforts have been made to develop unified visual language models capable of generating both text and visual content, including images and videos. There are two mainstream methods to generate visual content in VLMs. Many works (Sun et al., 2023b;a; Jin et al., 2023; Ge et al., 2023b; Li et al., 2023c; Ge et al., 2024; Jin et al., 2024; Ge et al., 2023a) combine VLMs with diffusion models like Stable Diffusion (Rombach et al., 2022a) for high-quality image generation. Other works (Liu et al., 2024a; Yu et al., 2023a; Lu et al., 2023; Team, 2024; Xie et al., 2024) adopt VQGAN-based vision encoders to convert visual inputs into discrete tokens and make LLMs learn to predict them. For more details on the distinction between our method and other unified visual language models, please refer to Appendix A.

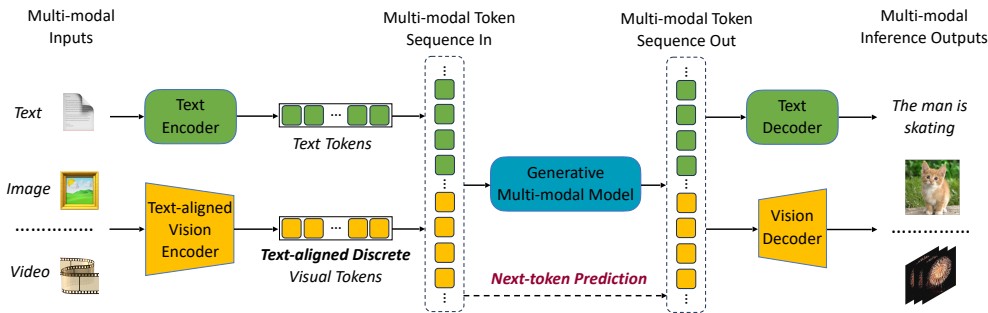

Figure 1: **An overview of our framework's multi-modal training and inference process.** Visual inputs are tokenized into discrete tokens and concatenated with textual tokens to form a multi-modal token sequence. All tokens are involved in our next-token prediction process, enabling a unified training objective. During inference, the output tokens are decoded by our text detokenizer or vision tower decoder to yield multi-modal content.

## 3 METHODS

This work proposes a multi-modal framework that aims to unify visual and language modalities effectively. The key components enabling such unification are a unified foundation vision tower that converts visual inputs into discrete tokens aligned with text, and a unified multi-modal generative training procedure. An overview of the main multi-modal training and inference process within our framework is depicted in Figure 1.

### 3.1 UNIFIED FOUNDATION VISION TOWER

To support diverse visual understanding and generation tasks, we first build a unified foundation vision tower to provide appropriate visual features. We propose to include text-image contrastive loss and VQ-based image reconstruction loss in our vision tower training, empowering the text alignment and discrete tokenization abilities for our vision tower. As depicted in Figure 2, the features extracted from images are primarily discretized through residual quantization. Then in one route, the discrete visual features are fed into a decoder to reconstruct the image and compute the reconstruction loss; on the other route, we compute the image-text contrastive loss between the discrete visual features and the textual features provided by a text encoder. With this training procedure, the vision tower learns to extract discrete features suitable for both understanding and generation in our VLM.

**Unified Training Recipe.** Training the unified vision tower with two objectives from scratch would be difficult, because alignment and reconstruction tasks require high-level semantic and low-level appearance features, respectively. Training the entire vision tower from scratch with both objectives could induce conflicting goals. In practice, we observe that training the vector-quantized vision tower from scratch with both image reconstruction and contrastive loss results in a mere 5% Top-1 accuracy for zero-shot image classification on ImageNet (Deng et al., 2009a) after several epochs of training.

To address this issue, we experiment with different training recipes (failed recipes are listed in Appendix C) and find the following solution to be most effective. Instead of learning both objectives simultaneously, we suggest first equipping the model with text-image alignment ability and then learning reconstruction while maintaining alignment ability. We initialize the vision encoder and text encoder with pretrained weights from the CLIP model to ensure good text-image alignment. Next, we freeze the text encoder and keep all vision components trainable using both contrastive and reconstruction loss. The contrastive loss maintains alignment ability, while the reconstruction loss develops reconstruction ability. This approach converges quickly and yields strong performance. The pre-trained CLIP weights contain learned high-level priors, which are difficult and computationally expensive to learn from scratch. Initializing with these weights enables the binding of low-level and high-level features much faster and more tractably for the vision encoder. With this recipe, we can train a vision tower that exhibits both good text alignment and image reconstruction abilities. We use weighted sum to combine the text-image contrastive loss and VQ-based image reconstruction loss:

$$\mathcal{L}_{total} = w_{contra}\mathcal{L}_{contra} + w_{recon}\mathcal{L}_{recon} \tag{1}$$

In our experiments, we pick $w_{contra} = 1$ and $w_{recon} = 1$.

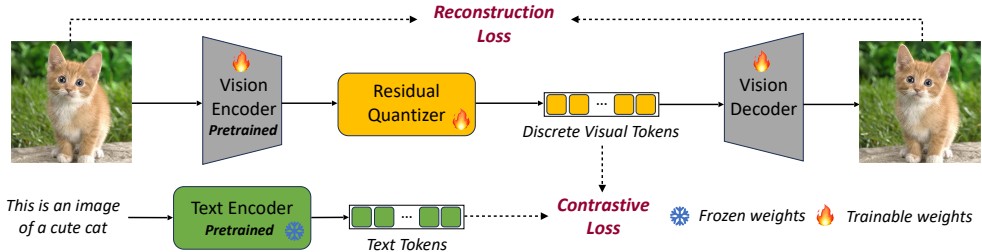

Figure 2: **Overview of our unified foundation vision tower.** Given input images the features extracted by the vision encoder are discretized using residual quantization. Then the discrete vision features are meanwhile put into the vision decoder to reconstruct images and used to perform the text-image alignment. During this process, the reconstruction loss and contrastive loss are computed to update the vision tower, endowing it to produce discrete visual features with text alignment.

**Residual Vector Quantization.** Our visual features are discretely quantized, so their representation ability heavily depends on the code size used in our quantizer. Since we hope they contain both high-level and low-level features, we need more capacities in their vector feature space, making a larger code size necessary for good performance in downstream tasks. However, too many codes for each image will result in too many tokens for LLM to produce in the visual generation process, incurring much latency. So in an attempt to increase the vector feature capacity and meanwhile maintain a reasonable number of tokens for LLM, we adopt a residual vector quantization method following RQ-VAE (Lee et al., 2022) to discretize a vector $\mathbf{z}$ as $D$ discrete codes:

$$\mathcal{RQ}(\mathbf{z}; \mathcal{C}, D) = (k_1, \cdots, k_D) \in [K]^D, \tag{2}$$

where $\mathcal{C}$ is the codebook, $K = |\mathcal{C}|$ and $k_d$ is the code of $\mathbf{z}$ at depth $d$. Starting with $\mathbf{r}_0 = \mathbf{z}$, we recursively perform vector quantization by

$$\begin{aligned} k_d &= \mathcal{Q}\left(\mathbf{r}_{d-1}, \mathcal{C}\right), \\ \mathbf{r}_d &= \mathbf{r}_{d-1} - \mathbf{e}\left(k_d\right), \end{aligned} \tag{3}$$

for each depth $d = 1, 2, \cdots, D$, where $\mathbf{e}$ is the codebook embedding table and $\mathcal{Q}$ is the standard vector quantization:

$$\mathcal{Q}(\mathbf{z}; \mathcal{C}) = \underset{k \in [K]}{\arg\min} \|\mathbf{z} - \mathbf{e}(k)\|_2^2. \tag{4}$$

The quantized vector for $\mathbf{z}$ is the sum over the depth dim: $\widehat{\mathbf{z}} = \sum_{i=1}^{D} \mathbf{e}\left(k_i\right)$. Intuitively, in each depth we choose a code to reduce the quantization error. So compared to the standard vector quantization methods, we have $D$ codes to quantize one vector, allowing for finer approximation and larger feature space. During multi-modal training and inference, LLM only needs to predict the code embedding, with codes in different depth sequentially produced by a depth transformer taking the code embedding as the initial input, as we will introduce in Section 3.2. So with this residual quantization, we can enhance the representation capability of our vision tower while incurring little latency.

## 3.2 UNIFIED MULTI-MODAL GENERATIVE PRE-TRAINING

Figure 1 presents an overview of our unified multi-modal pre-training process. Our vision tower encoder processes visual inputs sequentially, generating a 1D token sequence. This sequence is then concatenated with text tokens to form a multi-modal sequence. To distinguish between modalities and enable visual content generation, we insert special tokens: <image_start> and <image_end> at the start and end of image tokens, and <video_start> and <video_end> at the start and end of video tokens. Video tokens are the direct concatenation of multi-frame image tokens.

**Pre-training data form.** In terms of unified pre-training data, we leverage different concatenation forms between text and visual tokens to facilitate both understanding and generation. We use [image, text], [text, image], and [text, video] forms, with supervision loss added only on the latter modality in each pair to avoid unconditional content generation and promote modality alignment. We also employ an interleaved text and image concatenation form for enhanced understanding, with supervision loss applied solely to the text. Notably, we exclude the [video, text] form during pre-training for efficiency reasons, as we find incorporating it during supervised fine-tuning effectively yields excellent video understanding ability.

**Training Objective.** Since both visual tokens and text tokens are discrete, we can train our LLM with the general language modeling next-token prediction objective. However, due to the use of residual quantization for visual tokens, the training objectives for text and visual tokens differ slightly. For text tokens, the negative log-likelihood loss is calculated as

$$\mathcal{L}_{\text{text}} = -\sum_{i=1}^{T} \log P_\theta\left(y_i | y_{<i}\right),$$
(5)

where $T$ is the length of the multi-modal sequence and $i$ only counts when the text token appears at position $i$. For visual tokens, residual quantization introduces a depth-stacked structure of codes at each visual position $j$. To address this, we leverage the depth transformer introduced in RQ-VAE (Lee et al., 2022). Specifically, given the code embedding $h_j$ generated by the LLM for visual tokens at position $j$, the depth transformer autoregressively predicts D residual tokens $(k_{j1}, ..., k_{jD})$. During training, the input of the depth transformer $v_{jd}$ at depth d is defined as the sum of the code embeddings of up to depth $d-1$ for $d > 1$ such that

$$v_{jd} = \sum_{d'=1}^{d-1} \mathbf{e}(k_{jd'}),$$
(6)

and $v_{j1} = h_j$. Thus, the depth transformer predicts the next code for a finer estimation of the feature $\hat{z}_j$ based on the previous estimations up to $d-1$. Then the negative log-likelihood loss for visual tokens is

$$\mathcal{L}_{\text{visual}} = -\sum_{j=1}^{T} \sum_{d=1}^{D} \log P_\delta\left(k_{jd} | k_{j,<d}\right),$$
(7)

where $T$ is the length of the multi-modal sequence and $j$ only counts when a visual token appears at position $j$. During the multi-modal pre-training, the weights of the depth transformer are randomly initialized and updated together with the LLM.

## 4 EXPERIMENTS

In this section, we introduce comprehensive experiments to evaluate our method on various visual understanding and generation tasks. Firstly, we outline our experimental setup, including the model architecture, training datasets, and evaluation benchmarks. Subsequently, we evaluate the performance of our unified foundation vision tower. Then, we compare our method with other popular VLMs on various visual understanding and generation benchmarks. Finally, we give some qualitative results.

### 4.1 EXPERIMENTAL SETUP

In our experiments, we employ LLaMA-2-7B (Touvron et al., 2023b) as our base language model. For the vision tower, we choose SigLIP-Large-patch16-256 / SigLIP-SO400M-patch14-384 (Zhai et al., 2023) as our vision encoder architecture, and adopt the residual quantizer, depth transformer as well as the decoder architecture from RQ-VAE (Lee et al., 2022). The quantizer codebook size is 16384. All images and videos are resized to a resolution of $256 \times 256$ / $384 \times 384$, with each image or video frame converted into a $16 \times 16 \times 4$ / $27 \times 27 \times 16$ code with the residual depth $D = 4$ / $D = 16$. We train our vision tower on COYO-700M (Byeon et al., 2022) and evaluate it for zero-shot classification and reconstruction performance on ImageNet (Deng et al., 2009b). For visual understanding, we leverage 1M [image, text] data from ShareGPT4V (Chen et al., 2023), 6M interleaved text and image data from MMC4 (Zhu et al., 2024). For visual generation, we incorporate 15M high-quality [text, image] data curated from our internal dataset and 1M [text, video] data from OpenVid (Nan et al., 2024) datasets. Classifier-free guidance (Ho & Salimans, 2022) is employed for visual generation with a CFG value of 3.

For examining visual understanding ability, we evaluate our model on the widely adopted zero-shot image-based visual-language benchmarks including VQAv2 (Goyal et al., 2017), GQA (Hudson &

Manning, 2019), TextVQA (Singh et al., 2019), POPE (Li et al., 2023d), MME (Fu et al., 2024), SEED (Li et al., 2023a), MM-Vet (Yu et al., 2023b) and video-based visual-language benchmarks including ActivityNet (Caba Heilbron et al., 2015), MSVD (Chen & Dolan, 2011), MSRVTT (Xu et al., 2017), TGIF (Li et al., 2016).

To evaluate the visual generation capability, we use MJHQ-30K (Li et al., 2024) and GenAI-Bench (Lin et al., 2024) for image generation and VBench (Huang et al., 2024) for video generation. MJHQ-30K adopts the FID between generated images and 30K high-quality images to reflect the overall capability of image generation. GenAI-Bench is a challenging image-to-text generation benchmark that reflects the comprehensive generative abilities of image generation models. Vbench is a comprehensive benchmark suite for video generative models that decomposes the generation quality into multiple well-defined dimensions to facilitate fine-grained and objective evaluation.

## 4.2 UNIFIED FOUNDATION VISION TOWER

We present the commonly used metrics reconstruction FID (rFID) and Top-1 accuracy for zero-shot image classification on ImageNet to measure the reconstruction and text alignment capabilities of the unified foundation vision tower in Table 1. Please refer to the Appendix B.1 for the qualitative reconstruction results. Our model achieves significantly better reconstruction results than VQ-GAN. Our rFID is slightly inferior to that of RQ-VAE when using the same code shape. This is expected as the introduction of contrastive loss during training, aimed at enhancing image understanding, led to a decrease in reconstruction quality. For the text alignment capability, our unified vision tower achieves a Top-1 accuracy of 73.3 / 78.0 under 256 / 384 resolution. This demonstrates the exceptional text alignment capability of our unified vision tower. However, it is worth noting that both the rFID and Top-1 accuracy of the vision tower only serves as a medium indicator. As the unified vision tower is an integral component of the entire autoregressive model, we believe that its performance on downstream tasks, such as visual understanding and generation, holds greater significance.

Table 1: The reconstruction FID (rFID) and Top-1 accuracy for zero-shot image classification of our unified vision tower on ImageNet.

| Model | Pretrained Weights | Resolution | Shape of Code | rFID↓ | Top-1 Accuracy↑ |
|---|---|---|---|---|---|
| VQ-GAN | – | $256 \times 256$ | $16 \times 16$ | 4.98 | – |
| RQ-VAE | – | $256 \times 256$ | $8 \times 8 \times 4$ | 3.20 | – |
| RQ-VAE | – | $256 \times 256$ | $16 \times 16 \times 4$ | 1.30 | – |
| Ours | SigLIP-Large | $256 \times 256$ | $16 \times 16 \times 4$ | 1.80 | 73.3 |
| Ours | SigLIP-SO400M | $384 \times 384$ | $27 \times 27 \times 16$ | 1.25 | 78.0 |

## 4.3 QUANTITATIVE EVALUATION

**Visual Understanding Tasks.** Table 2 and Table 3 summarize the comparison between our method and other leading VLMs on the image-language and video-language benchmarks respectively. Compared to the mainstream choice of continuous visual tokens produced by foundation models like CLIP, the VQGAN-based discrete visual tokens have less alignment with text, thus harming VLMs' performance on visual understanding tasks. With our unified foundation vision tower, our model can have a performance close to leading VLMs even with discrete visual tokens.

**Visual Generation Tasks.** As shown in Table 4, VILA-U can achieve a better FID than other autoregressive methods and have comparable performance with some diffusion based methods. This result shows the feasibility of our method for visual generation. Table 5 summarizes the quantitative results of our method and other visual generation methods on GenAI-Bench. Although Our method is inferior to diffusion-based visual generation methods that have been trained on billions-level image-text pairs,

| Method | Type | #Images | FID↓ |
|---|---|---|---|
| SD v2.1 | Diffusion | – | 26.96 |
| SD-XL | Diffusion | 2000M | 9.55 |
| PixArt | Diffusion | 25M | 6.14 |
| Playground v2.5 | Diffusion | – | 4.48 |
| LWM | Autoregressive | – | 17.77 |
| Show-o | Autoregressive | 36M | 15.18 |
| Ours (256) | Autoregressive | 15M | 12.81 |
| Ours (384) | Autoregressive | 15M | 7.69 |

Table 4: Comparison with other visual generation methods on MJHQ-30K evaluation benchmark.

Table 2: Comparison with leading methods on image-based visual language benchmarks. Our performance is close to leading VLMs, surpassing many methods by a large margin under the same LLM size, even with a discrete visual token type. * indicates that images in the training split of these datasets are observed during VLM training.

| Method | LLM | Visual Token | Res. | VQAv2 | GQA | TextVQA | POPE | MME | SEED | MM-Vet |
|--------|-----|-------------|------|-------|-----|---------|------|-----|------|--------|
| LLaVA-1.5 | Vicuna-1.5-7B | Continuous | 336 | 78.5* | 62.0* | 58.2 | 85.9 | 1510.7 | 58.6 | 30.5 |
| VILA | LLaMA-2-7B | Continuous | 336 | 79.9* | 62.3* | 64.4 | 85.5 | 1533.0 | 61.1 | 34.9 |
| Unified-IO 2 | 6.8B from scratch | Continuous | 384 | 79.4* | – | – | 87.7 | – | 61.8 | – |
| InstructBLIP | Vicuna-7B | Continuous | 224 | – | 49.2 | 50.1 | – | – | 53.4 | 26.2 |
| IDEFICS-9B | LLaMA-7B | Continuous | 224 | 50.9 | 38.4 | 25.9 | – | – | – | – |
| Emu | LLaMA-13B | Continuous | 224 | 52.0 | – | – | – | – | – | – |
| LaVIT | LLaMA-7B | Continuous | 224 | 66.0 | 46.8 | – | – | – | – | – |
| DreamLLM | Vicuna-7B | Continuous | 224 | 72.9* | – | 41.8 | – | – | – | 36.6 |
| Video-LaVIT | LLaMA-2-7B | Continuous | 224 | 80.2* | 63.6* | – | – | 1581.5 | 64.4 | 35.0 |
| Emu2-Chat | Emu2-37B | Continuous | 448 | 84.9* | 65.1* | 66.6* | – | – | – | – |
| MM-Interleaved | Vicuna-13B | Continuous | 224 | 80.2* | 60.5* | 61.0 | – | – | – | – |
| DEEM | Vicuna-7B | Continuous | 448 | 68.2* | 55.7* | – | – | – | – | 37.4 |
| CM3Leon-7B | 7B from scratch | Discrete | 256 | 47.6 | – | – | – | – | – | – |
| LWM | LLaMA-2-7B | Discrete | 256 | 55.8 | 44.8 | 18.8 | 75.2 | – | – | 9.6 |
| Show-o | Phi-1.5-1.3B | Discrete | 256 | 59.3* | 48.7* | – | 73.8 | 948.4 | – | – |
| SEED-LLaMA | Vicuna-7B | Discrete | 224 | 66.2 | – | – | – | – | 51.5 | – |
| Ours | LLaMA-2-7B | Discrete | 256 | 75.3* | 58.3* | 48.3 | 83.9 | 1336.2 | 56.3 | 27.7 |
| Ours | LLaMA-2-7B | Discrete | 384 | 79.4* | 60.8* | 60.8 | 85.8 | 1401.8 | 59.0 | 33.5 |

Table 3: Comparison with leading methods on video-based visual language benchmarks. The performance of our method is close to state-of-the-art VLMs, surpassing many methods under the same LLM size, even with a discrete visual token type.

| Method | LLM | Visual Token | Res. | MSVD-QA | MSRVTT-QA | TGIF-QA | Activity Net-QA |
|--------|-----|-------------|------|---------|-----------|---------|-----------------|
| Unified-IO 2 | 6.8B from scratch | Continuous | 384 | 52.1 | 42.5 | – | – |
| Emu | LLaMA-13B | Continuous | 224 | – | 18.8 | 8.3 | – |
| VideoChat | Vicuna-7B | Continuous | 224 | 56.3 | 45 | 34.4 | – |
| Video-LLaMA | LLaMA-2-7B | Continuous | 224 | 51.6 | 29.6 | – | – |
| Video-ChatGPT | LLaMA-2-7B | Continuous | 224 | 64.9 | 49.3 | 51.4 | 35.2 |
| Video-LLava | Vicuna-7B | Continuous | 224 | 70.7 | 59.2 | 70.0 | 45.3 |
| Video-LaVIT | LLaMA-2-7B | Continuous | 224 | 73.5 | 59.5 | – | 50.2 |
| Emu2-Chat | Emu2-37B | Continuous | 448 | 49.0 | 31.4 | – | – |
| LWM | LLaMA-2-7B | Discrete | 256 | 55.9 | 44.1 | 40.9 | – |
| SEED-LLaMA | Vicuna-7B | Discrete | 224 | 40.9 | 30.8 | – | – |
| Ours | LLaMA-2-7B | Discrete | 256 | 73.4 | 58.9 | 51.3 | 51.6 |
| Ours | LLaMA-2-7B | Discrete | 384 | 75.3 | 60.0 | 51.9 | 52.7 |

our method has comparable performance with SD v2.1 (Rombach et al., 2022b) and SD-XL (Podell et al., 2023) on *advanced* prompts even trained with magnitude-level less data. This further shows that VILA-U can learn the correlation among visual and textual modalities effectively with our unified training framework. For video generation, we evaluate our method on VBench (Huang et al., 2024) and compare it against Open-Sora (Zheng et al.), CogVideo (Hong et al., 2022), and CogVideoX (Yang et al., 2024). The results, presented in Table 6, demonstrate that our method achieves performance that is better than CogVideo and comparable to Open-Sora, highlighting the effectiveness of our approach.

## 4.4 QUALITATIVE EVALUATION

**Visual Understanding.** To validate the effectiveness of VILA-U in comprehensive visual understanding tasks, we apply it in several understanding and reasoning tasks, as some examples shown in Figure 3 and Figure 4. From the results, we can see the versatility of VILA-U in various tasks including visual captioning and visual question answering. Besides, our model has inherited some important capabilities from VILA (Lin et al.,

| Method | Total Score↑ | Quality Score↑ | Semantic Score↑ |
|--------|-------------|----------------|-----------------|
| Open-Sora | 75.91 | 78.82 | 64.28 |
| CogVideo | 67.01 | 72.06 | 46.83 |
| CogVideoX | 81.61 | 82.75 | 77.04 |
| Ours (256) | 74.01 | 76.26 | 65.04 |

Table 6: Comparison with other visual generation methods on VBench (Huang et al., 2024).

Table 5: Comparison with other visual generation methods on GenAI-Bench (Lin et al., 2024). The results show that our method outperforms previous autoregressive visual generation methods. For *advanced* prompts that require better text following ability to generate, our method can have a relatively small performance gap with diffusion-based methods, even with much less training data.

| Method | Type | #Training Images | Attribute↑ | Scene↑ | Relation↑ | | | Overall↑ |
| --- | --- | --- | --- | --- | --- | --- | --- | --- |
| | | | | | Spatial | Action | Part | |
| SD v2.1 | Diffusion | 2000M | 0.80 | 0.79 | 0.76 | 0.77 | 0.80 | 0.78 |
| SD-XL | Diffusion | 2000M | 0.84 | 0.84 | 0.82 | 0.83 | 0.89 | 0.83 |
| Midjourney v6 | Diffusion | – | 0.88 | 0.87 | 0.87 | 0.87 | 0.91 | 0.87 |
| DALL-E 3 | Diffusion | – | 0.91 | 0.90 | 0.92 | 0.89 | 0.91 | 0.90 |
| LWM | Autoregressive | – | 0.63 | 0.62 | 0.65 | 0.63 | 0.70 | 0.63 |
| Show-o | Autoregressive | 36M | 0.72 | 0.72 | 0.70 | 0.70 | 0.75 | 0.70 |
| Ours (256) | Autoregressive | 15M | 0.78 | 0.78 | 0.77 | 0.78 | 0.79 | 0.76 |
| Ours (384) | Autoregressive | 15M | 0.75 | 0.76 | 0.75 | 0.73 | 0.75 | 0.73 |

(a) VQAScores on *basic* prompts of GenAI-Bench

| Method | Type | #Training Images | Count↑ | Differ↑ | Compare↑ | Logical↑ | | Overall↑ |
| --- | --- | --- | --- | --- | --- | --- | --- | --- |
| | | | | | | Negate | Universal | |
| SD v2.1 | Diffusion | 2000M | 0.68 | 0.70 | 0.68 | 0.54 | 0.64 | 0.62 |
| SD-XL | Diffusion | 2000M | 0.71 | 0.73 | 0.69 | 0.50 | 0.66 | 0.63 |
| Midjourney v6 | Diffusion | – | 0.78 | 0.78 | 0.79 | 0.50 | 0.76 | 0.69 |
| DALL-E 3 | Diffusion | – | 0.82 | 0.78 | 0.82 | 0.48 | 0.80 | 0.70 |
| LWM | Autoregressive | – | 0.59 | 0.58 | 0.54 | 0.49 | 0.52 | 0.53 |
| Show-o | Autoregressive | 36M | 0.70 | 0.62 | 0.71 | 0.51 | 0.65 | 0.60 |
| Ours (256) | Autoregressive | 15M | 0.70 | 0.71 | 0.74 | 0.53 | 0.66 | 0.64 |
| Ours (384) | Autoregressive | 15M | 0.68 | 0.67 | 0.71 | 0.51 | 0.64 | 0.61 |

(b) VQAScores on *advanced* prompts of GenAI-Bench

2023) including multi-image understanding, in-context learning, as shown in Figure 5 and Figure 6. More visualizations can be found in the Appendix B.2 and B.3.

**Visual Generation.** We present some examples of the visual generation results in Figure 7. Our model can be employed in both image generation and video generation, even trained with a relatively small data corpus. In the given examples, our method can generate nice-looking images and continuous videos adhering to the user's input. More visualizations can be found in the appendix B.4.

## 5  ABLATION STUDY

### 5.1  IMPACT OF CONTRASTIVE LOSS TO VISUAL UNDERSTANDING

We include contrastive loss in vision tower training, which endows it with the text alignment ability. During our multi-modal training, such text alignment ability is crucial in enhancing modality fusion and performance on downstream visual language tasks. We validate the importance of this alignment by training the vision tower with and without the contrastive loss, evaluating its impact on visual language understanding performance. For this ablation study, we randomly sample 25M data from

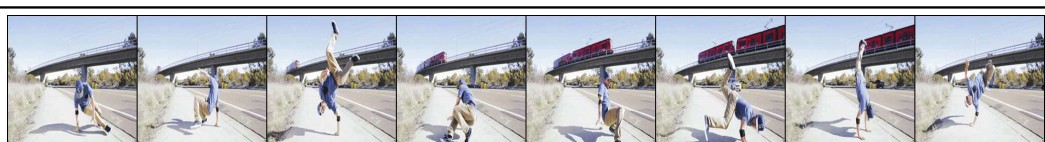

**Prompt:** `<video>\n`
Elaborate on the visual and narrative elements of the video in detail.
**VILA-U(Ours):** The video features a person wearing a blue t-shirt, khaki pants, and a cap, executing a series of breakdancing moves on a concrete road. The background includes a clear sky, a bridge, and a train passing by. The dancer's movements are dynamic, with spins, flips, and handstands. The lighting is bright, suggesting daytime, and the shadows indicate the sun is high. The dancer's attire is casual, and the setting appears to be a quiet road with minimal traffic.

Figure 3: VILA-U can correctly caption videos and cover all the details, thanks to the text alignment of our vision encoder.

| Input image | Question | Output |
|---|---|---|
| | Is there water visible? | Yes |
| | What are flying in the sky? | Kites |
| | How many motorcycles are in the picture? | 2 |
| | How many people are on each motorcycle? | 0 |

Figure 4: VILA-U has good visual question answering capability. The images and questions are from the test split of VQAv2 dataset.

| Input few shot examples + target image | Output |
|---|---|
| is home to polar bears    is home to pandas    ? | is home to camels |
| is sunny    is rainy    ? | is snowy |

Figure 5: VILA-U has good in-context learning capability. We feed two image-text pairs and a third image as the context to prompt the VLM.

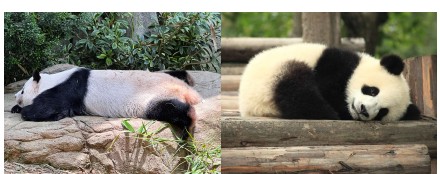

**Prompt:** Image 1: `<img1>`\n Image 2: `<img2>`\n
What is the common thing and the difference between image 1 and image 2?
**VILA-U(Ours):** The common thing in both images is the presence of a panda bear. However, the difference between image 1 and image 2 is that the panda bear in image 1 is lying down on a rock, while the panda bear in image 2 is lying down on a log.

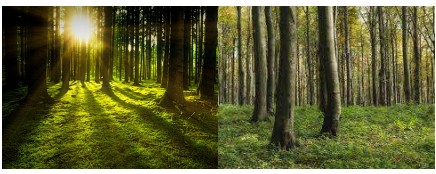

**Prompt:** Image 1: `<img1>`\n Image 2: `<img2>`\n
What is the common thing and the difference between image 1 and image 2?
**VILA-U(Ours):** The common thing in both images is the presence of trees, but the difference between the two images is the angle of the sunlight. In image 1, the sun is shining through the trees, creating a bright and lively atmosphere. In contrast, the sun is not visible in image 2, which shows a darker and more subdued forest scene.

Figure 6: VILA-U can correctly reason over multiple images.

COYO-700M to train the vision tower. For multi-modal training, we use ShareGPT4V and MMC4 without text-image and text-video data. The results of the first two lines in Table 7 demonstrate the crucial role of text alignment in achieving strong visual language understanding performance. Scaling the dataset size from 25M to 700M further enhances performance, highlighting the importance of learning text alignment on a large-scale dataset.

Table 7: Impact of contrastive loss to visual understanding.

| Pretrained Weights | Data size | Loss Type | Top-1 Accuracy | VQAv2 | POPE | MME | SEED | MM-Vet |
|---|---|---|---|---|---|---|---|---|
| SigLIP-Large | 25M | Recon. | – | 57.7 | 75.1 | 937.7 | 38.7 | 15.3 |
| SigLIP-Large | 25M | Recon. + Contra. | 62.9 | 68.0 | 83.7 | 1219 | 50.4 | 20.8 |
| SigLIP-Large | 700M | Recon. + Contra. | 73.3 | 75.3 | 83.9 | 1336.2 | 56.3 | 27.7 |

## 5.2 IMPACT OF CONTRASTIVE LOSS TO VISUAL GENERATION

We conduct two experiments to demonstrate the influence of contrastive loss to generation performance. For efficiency, we conduct only text-to-image pretraining and utilize Sheared-LLaMA-1.3B (Xia et al., 2023) instead of LLaMA-2-7B as the LLM. In the first experiment, we use the RQ-VAE as the vision tower, which has an rFID of 1.30. In the second experiment, we employ our unified vision tower. Results are shown in Table 8. Our Unified Vision Tower yielded slightly worse FID results than the RQ-VAE on MJHQ-30K, possibly due to its inferior rFID resulting from the contrastive loss.

Table 8: Impact of contrastive loss to visual generation.

| Vision Tower | LLM | Resolution | rFID ↓ | FID ↓ |
|---|---|---|---|---|
| RQ-VAE (Lee et al., 2022) | Sheared-LLaMA-1.3B | 256 × 256 | 1.30 | 12.0 |
| Ours | Sheared-LLaMA-1.3B | 256 × 256 | 1.80 | 13.2 |

Table 9: Impact of CFG.

| CFG Value | FID ↓ |
|---|---|
| 1.0 | 14.1 |
| 2.0 | 13.0 |
| 3.0 | 12.8 |
| 5.0 | 13.2 |

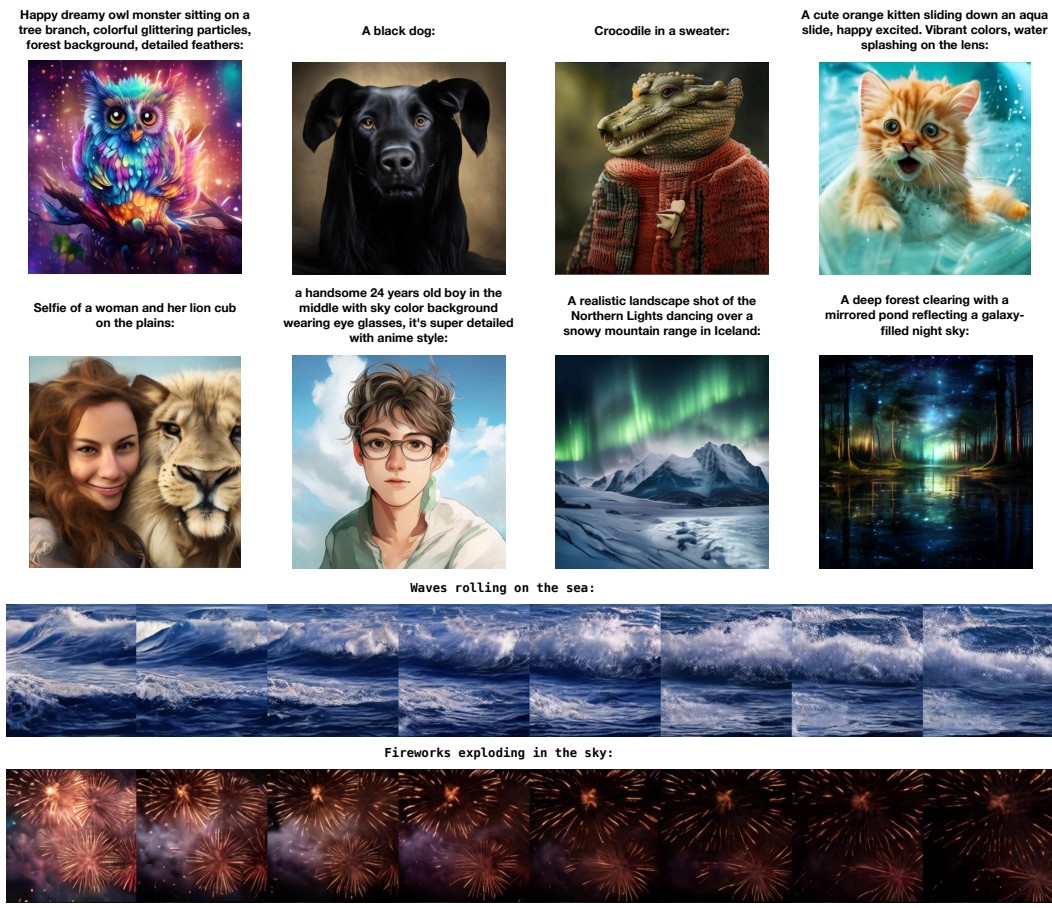

Figure 7: VILA-U can generate high-quality images and videos given text input.

## 5.3 Impact of Classifier-free Guidance

We adopt classifier-free guidance during the visual content generation. We investigate the impact of the CFG value on our 256-resolution model. Results presented in Table 9 indicate that a CFG value of 3.0 yields the best FID score.

## 6 Conclusion and Limitation

We present VILA-U, a novel and unified visual language model that integrates video, image and language understanding and generation tasks into one autoregressive next-token prediction framework. Our method is not only more concise than most VLMs that leverage additional components like diffusion models for unifying visual generation and understanding, but also demonstrates that autoregressive methods can achieve comparable performance to state-of-the-art VLMs. We believe VILA-U can serve as a general-purpose framework for diverse visual language tasks.

As demonstrated in Section 5.2, the introduction of contrastive loss impacts the reconstruction ability of the vision tower. Balancing these two capabilities within the unified vision tower presents an interesting and complex challenge that requires further exploration. Additionally, we currently do not observe significant synergy or mutual enhancement between understanding and generation tasks. In the future, we aim to investigate and explore more effective methods to enable these tasks to complement and reinforce each other, thereby fully realizing the untapped potential of a unified visual language model.

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

## APPENDIX

## A    DIFFERENCE WITH RELATED WORKS

Prior to VILA-U, unified visual language models were dominated by two mainstream approaches:

(1) Represented by LWM, CM3Leon and Show-o which utilizes a VQGAN-based tokenizer to convert visual inputs into discrete tokens. However, as these tokenizers are trained solely with a reconstruction objective, the resulting tokens lack rich semantic information. This limitation leads to poor performance on multimodal understanding tasks. But it can easily support autoregressive visual generation and the generated visual tokens can be seamlessly decoded into visual outputs using the lightweight decoder of VQGAN.

(2) Represented by AnyGPT SEED-LLaMa and LaViT, which utilizes a codebook to quantize features produced by a pre-trained ViT model like CLIP. Since CLIP features encode rich semantic information, these approaches generally achieve significantly better performance on understanding tasks compared to VQGAN-based tokenizers. However, these tokenizers lack decoding capability, requiring an external visual generation model, such as a diffusion model, to use the generated visual tokens as conditions for producing visual outputs.

Compared to these two mainstream approaches, VILA-U introduces a solution that addresses the limitations of both. We design a unified vision tower that extracts features with rich semantic information, similar to CLIP, while also supporting image reconstruction capabilities akin to VQGAN. This is achieved by incorporating both reconstruction loss and contrastive loss into the autoencoder training process, along with utilizing residual quantization to enhance the representation capability of the visual features. Building on this foundation, we develop a single end-to-end autoregressive framework that eliminates the need for external visual generation models required by approach 2 and significantly outperforms the understanding results of methods in approach 1.

## B    QUALITATIVE RESULTS

### B.1    RECONSTRUCTION

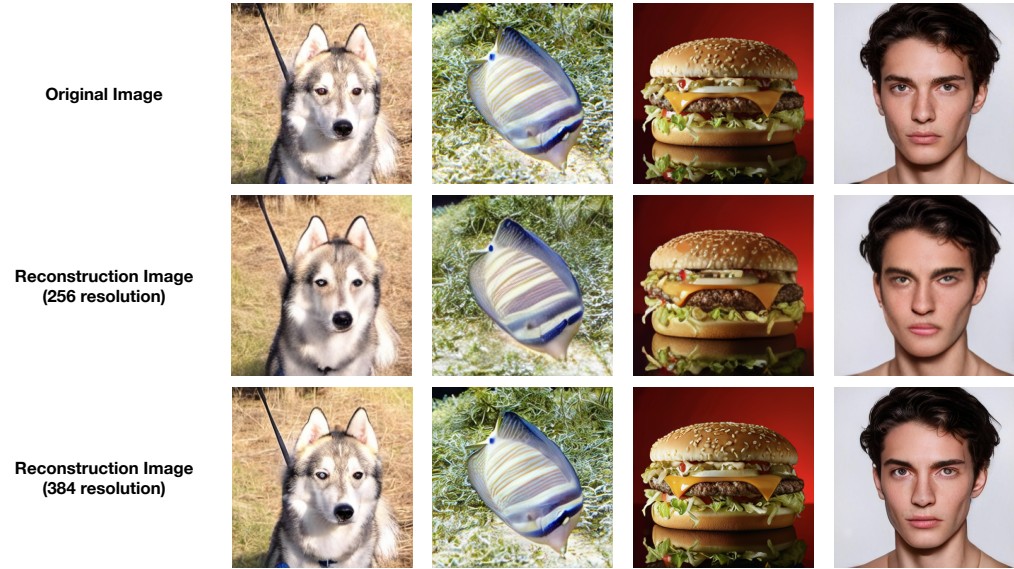

Figure 8: Visualization of the reconstruction results from text-aligned discrete visual tokens.

We present qualitative reconstruction results in Figure 8 for our 256 / 384 resolution vision tower. These vision towers effectively reconstruct images in detail using text-aligned discrete visual tokens.

## B.2 VISUAL UNDERSTANDING

| Input image | Question | Output |
|---|---|---|
| | What animals are shown here? | Zebras |
| | What color are they? | Black and white |
| | What is the color of the far right board? | Blue |
| | How many men are in the picture? | 2 |
| | Is she cooking bread? | Yes |
| | What room is this? | Kitchen |
| | Did the man hit the bat? | No |
| | Is the man's legs together or apart? | Apart |

Figure 9: Image understanding results. Examples are taken from the test split of VQAv2 dataset.

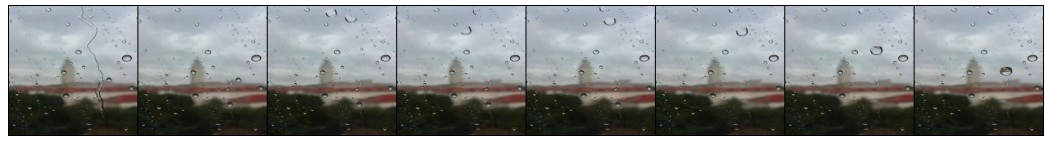

**Prompt:** `<video>\n` What do the drops do 5 times ?
**VILA-U(Ours):** The drops fall down the window 5 times, and each time they hit the window, they create a new set of drops.

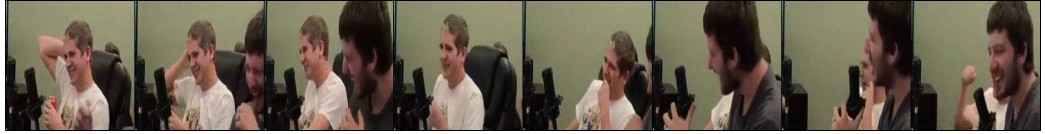

**Prompt:** `<video>\n` how many guys are laughing ?
**VILA-U(Ours):** 2 guys are laughing.

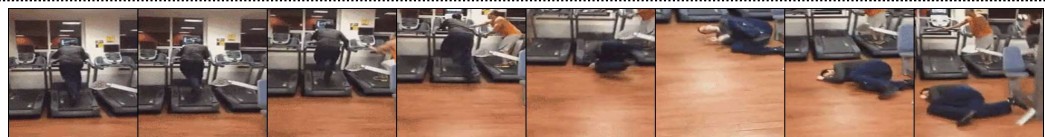

**Prompt:** `<video>\n` where is the man running and then he fall ?
**VILA-U(Ours):** The man is running on a treadmill in a gym, and then he falls off the treadmill.

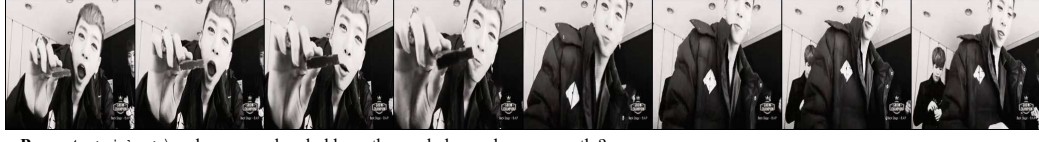

**Prompt:** `<video>\n` how many boy holds up the candy bar and opens mouth ?
**VILA-U(Ours):** The video shows one boy holding up the candy bar and opening his mouth.

Figure 10: Video understanding results. Examples are taken from the test split of TGIF dataset.

We provide more image understanding and video understanding examples in Figure 9 and Figure 10. VILA-U successfully answers the questions accurately.

## B.3  In-context Learning Examples

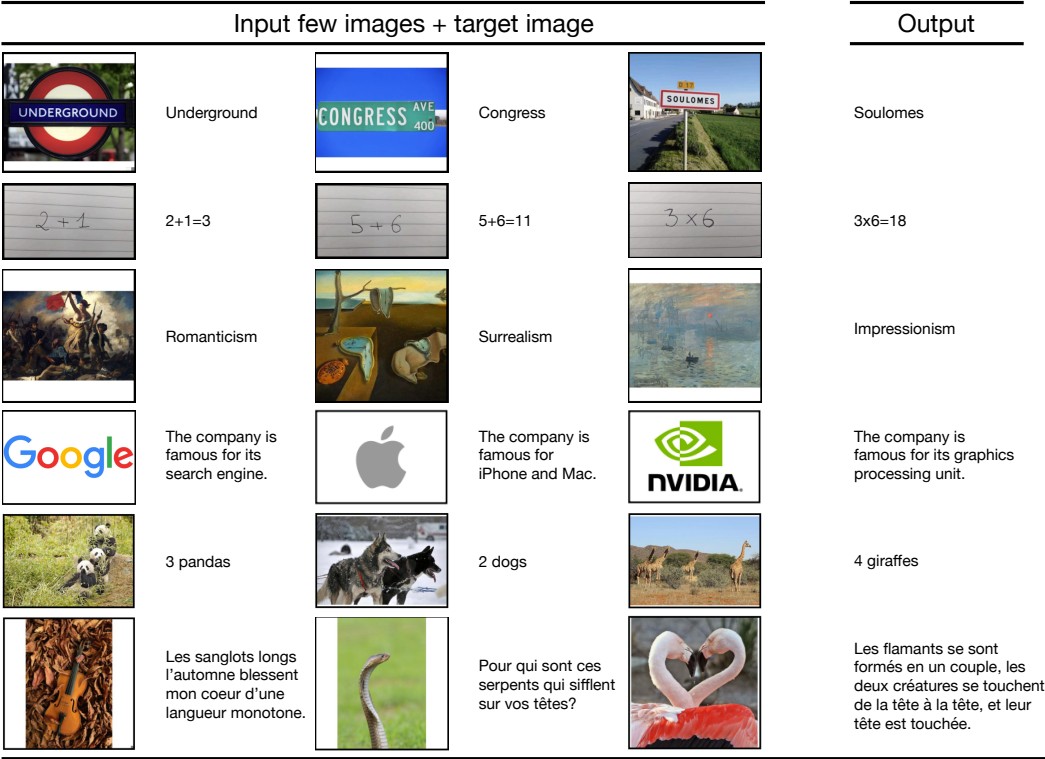

Figure 11: In-context learning examples. We try all in-context learning examples in Lin et al. (2023). The results demonstrate that VILA-U has inherited good in-context learning capabilties.

We provide more qualitative results to demonstrate in-context learning capabilities of VILA-U in Figure 11. VILA-U exhibits good in-context learning capabilties.

## B.4  Visual Generation

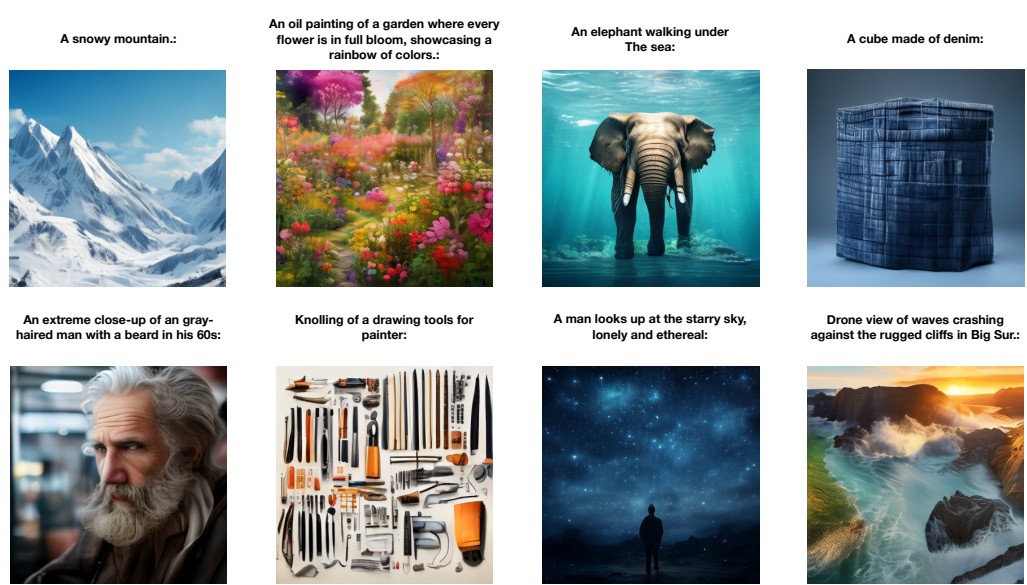

Figure 12: Image generation results. VILA-U can generate high-quality images given text input.

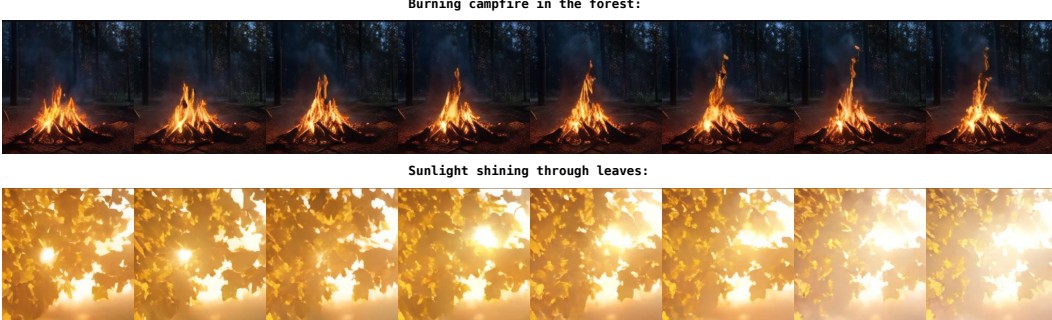

Figure 13: Video generation results. VILA-U can generate high-quality videos given text input.

We provide more image generation and video generation examples in Figure 12 and Figure 13. VILA-U can generate high-quality images and videos given text input.

## C  FAILED TRAINING RECIPES.

We experiment with numerous training recipes and find none to be as effective as our final approach. We list four alternative recipes and discuss their shortcomings compared to our final recipe: 1) Load pre-trained CLIP weights into the text encoder only; 2) Load pre-trained RQ-VAE weights for the vision encoder and decoder while training other parts from scratch; 3) Freeze the vision encoder; 4) Make the text encoder trainable.

Recipes 1) and 2) fail due to the lack of pre-trained CLIP weights for the vision encoder. Training a CLIP model from scratch typically requires numerous GPU days with a large global batch size (e.g., 32k). However, VQ-based reconstruction training necessitates a relatively small global batch size (e.g., 512) for steady improvement. With such a small batch size, training a text-aligned vision tower from scratch would be prohibitively time-consuming and resource-intensive.

Recipe 3) fails because freezing the vision encoder prevents it from learning the low-level features essential for reconstruction. In this case, the burden of reconstruction falls entirely on the vision decoder, but it is impossible to reconstruct images well using only semantic features.

Recipe 4) fails because the quantized features are chaotic during the initial training steps, and the contrastive loss disrupts the text encoder weights, slowing down the entire training process.

In contrast, our final training recipe leverages pre-trained CLIP weights for the vision encoder, enabling it to maintain learned semantic features rather than grasping them from scratch. This allows us to train with a small batch size while keeping the vision encoder trainable, facilitating the learning of low-level features for reconstruction during training.

