# OpenReview forum: "VILA-U: a Unified Foundation Model Integrating Visual Understanding and Generation"
_ICLR.cc/2025/Conference — ICLR 2025 Poster_

### Official Review · Reviewer_n4tc · 2024-10-28

**Soundness:** 3
**Presentation:** 3
**Contribution:** 3
**Rating:** 6
**Confidence:** 5

**Summary:**

Summary:

VILA-U is a foundation model that unifies video, image, and language understanding and generation. Unlike traditional models that use separate components for different tasks, VILA-U simplifies this by employing a single autoregressive framework. This reduces misalignment and maintains near state-of-the-art performance in both understanding and generating visual language content. Key factors for its success include a unified vision tower that aligns visual and textual inputs, enhancing perception, and the ability to achieve high-quality image generation similar to diffusion models.

Contributions:

1.  VILA-U strives for an end-to-end autoregressive model that handles both visual and textual inputs through a unified next-token prediction approach. This approach eliminates the need for external components like diffusion models, simplifying the infrastructure.
2.  VILA-U is tested across a range of tasks, including image-language and video-language understanding, as well as image and video generation. It demonstrates notable improvements, particularly in narrowing the gap between autoregressive and continuous-token models in visual understanding, while also offering robust visual generation capabilities.

**Strengths:**

1. The idea of VILA-U is very straightforward, and the experiments are solid. It significantly enhances the capabilities of end-to-end autoregressive multimodal models in visual-language tasks, bridging the gap between autoregressive multimodal models and the LLAVA series, while also excelling in image generation.

2. The structure of the VILA-U paper is simple and easy to read, and the model implementation is very easy to follow.

**Weaknesses:**

1.Regarding the issue of missing in context learning assessments, VILA-U has undergone extensive training on image-text sequences and can accept any interleaved layouts of images and text. Therefore, it should possess excellent contextual learning abilities. This work could be enhanced by conducting tests on its ICT capabilities.

2.The description of the data curation process is not sufficiently clear, making it uncertain whether the data was meticulously selected or randomly chosen. If it is the former, I suspect that most of the improvements stem from high-quality data engineering rather than advancements in model architecture.

**Questions:**

1. The  solid experimental results of VILA-U have largely reignited my confidence in the autoregressive image-text unified modeling direction. However, why is there no comparison with other text-image unified modeling models such as \textbf{MM-Interleaved, SEED, and DEEM} on image understanding tasks? Ignoring the contributions of pioneers is not advisable.

2. The video generation experiments are insufficient. Why not compare with methods like \textbf{OpenSora} and \textbf{CogVideoX} on \textbf{VBench}?

3. The article is unclear in its expression; are the visual tokens features directly discretized by the visual encoder, or are they encoded by a large language model? I suspect it is the former.

4. VILA-U claims to have lower computational complexity and to avoid misalignment. While I recognize the importance of addressing misalignment, the claim of lower complexity requires experimental support. Specifically, compared to unified autoregressive image-text modeling models, using separate models like fine-tuning Stable Diffusion can also construct end-to-end autoregressive image-text modeling, which is more efficient in training and performs better. Moreover, utilizing existing mature acceleration schemes offers fast speeds. VILA-U should emphasize more on data cleansing quality and misalignment.

5. Lastly, and most critically, I hypothesize that the structural improvements of the model provide minimal benefits compared to previous autoregressive unified models, with the majority of improvements stemming from the engineered data cleansing. For instance, MMC4-Core contains 22.4M data while MMC4 has 375M, yet some research indicates that training with these two datasets yields similar outcomes. Large-scale datasets like MMC4 are of very low quality. However, using just 6M of data to achieve excellent results suggests that your data is meticulously filtered, yet the paper lacks any detail on the core contributions of data construction. Conducting experiments on the same data with other model structures like \textbf{DreamLLM} is necessary to demonstrate the efficiency of \textbf{VILA-U}.

I will improve my rating score if my concerns are addressed.

**Details Of Ethics Concerns:**

All datasets used are public, no ethics review needed.

---

> ### Comment · Reviewer_n4tc · 2024-11-25
>
> Sorry for the late reply. Thank the author for the detailed rebuttals. My main concerns have been addressed, so I increase my score to 6. Looking forward for you open-sourced codebase and models. Please add missing references about DEEM, SEED, and MM-Interleaved.

---

### Official Review · Reviewer_X72f · 2024-10-30

**Soundness:** 2
**Presentation:** 3
**Contribution:** 3
**Rating:** 6
**Confidence:** 4

**Summary:**

The paper, VILA-U presents a unified framework of autoregressive multimodal generation and understanding. It achieves this by first training a vision encoder (discretized via RQ codebook) for text-conditioned image tokens (initialized from CLIP) and then training image+text data using autoregressive modeling. It presents a complete training recipe for creating autoregressive multimodal models, and the resulting model is benchmarked against a wide range of existing models across tasks (generation and understanding)

**Strengths:**

1. The unification of multiple modalities in the same architecture (with the same training objective) is a very important topic. The paper is a valuable contribution to this overall research program. In the current work, the choice of quantized image tokens for image representation makes the autoregressive modeling task more natural as the image modality is tokenized into discrete tokens much like language. This helps minimizes the amount of code development required for adapting existing LLM code bases to their multimodal counterparts.
2. The paper performed fairly complete evaluations (image-text, video-text, text-image, ) and ablation studies that include model backbone and training objective.

**Weaknesses:**

1. It is not clear to me how to position the work in its novelty or effectiveness and this may be addressable with some rewriting. I see 3 potential angles
    1. Training effectiveness by leveraging pretrained networks. The authors motivates the work by emphasizing that existing methods that attempt to unify multimodal generation and understanding either require significant architectural modifications to their uni-modal counterparts, or training from scratch. However, this comparison seems not to play a central role in the subsequent discussions. If the effectiveness of the proposed method is reflected in ease of training, then readers would expect to see comparison of training time/compute for comparable performances.
    2. Effective token representation of image modality as discrete tokens: VILA-U differs from prior work in its adoption of RQ-VAE embedding for images. However, if this is the main innovation, the choice of RQ, its superiority over alternative methods, the important of discontinuous embedding of images (as compared to, for example, continuous embedding as in LaViT) will need to be elevated.
    3. State-of-the-art performance: If the main contribution is instead just the shear effectiveness of the method. Then it should demonstrate this quantitative in the paper. Unfortunately, the comparison tables doesn’t seem to suggest that the VILA-U model is the state-of-the-art in most benchmarks. Perhaps it achieves Pareto frontier between understanding and generation tasks? Or outperforms other models for the same training compute/time? Either way I’m not clear what the main advantage of the current work is over others.
2. The discussion around training recipe is very important and useful for practitioners. However, it lacks both quantitative and qualitative (with examples) comparisons of the different training recipes. With the conclusion seems to be use an aligned CLIP model for image encoder initialization, which doesn’t seem to be a novel finding. I would recommend either supporting the discussion with more evaluation (quantitive or qualitative, ideally both) or moving the discussion to the appendix.
3. The paper suffers from unsubstantiated claims ( neither references nor experimental support). I've highlighted a few statements that are very important for the message in the paper below:
    - "replacing continuous tokens with VQ tokens in VLMs usually results in a severe performance drop"
    - "A straightforward combination of contrastive and reconstruction loss cannot converge"
    - "both the rFID and Top-1 accuracy of the vision tower only serves as a medium indicator instead of directly linearly correlated to the final performance of our whole multi-modal framework."

**Questions:**

My biggest suggestion/question is related to the number 1 weakness described above. If the author could highlight the main contribution of the work that would make its positioning much easier. One positioning that was left out in the weakness section above is to position the work as the "first" in some regards. However, while autoregressive modeling of text + language is a burgeoning field, VILA-U is not the first model that performs autoregressive modeling of multiple modalities.

---

### Official Review · Reviewer_ma7u · 2024-11-02

**Soundness:** 3
**Presentation:** 3
**Contribution:** 3
**Rating:** 6
**Confidence:** 4

**Summary:**

The paper presents VILA-U, a unified foundation model for visual understanding and generation that integrates image and language processing into a single autoregressive next-token prediction framework. Unlike traditional visual language models that rely on separate modules or diffusion models for generation, VILA-U employs a unified vision tower to discretize visual inputs, aligning them with textual tokens through contrastive learning. From the experiments, the authors show that VILA-U can achieve state-of-the-art performance in both image generation and comprehension.

**Strengths:**

1. VILA-U introduces a unified framework that handles both visual understanding and generation in a single autoregressive next-token prediction model.

2. The model leverages a unified vision tower that uses contrastive learning to align discrete visual tokens with textual inputs, which enhances the model's visual perception and text-visual alignment capabilities.

3. The experiments indicate the state-of-the-art performance of VILA-U in both image generation and understanding.

**Weaknesses:**

1. Missing the clarification between VILA-U and other tokenization-based multimodal models, like AnyGPT [1] and SEED-LLaMa [2]. Those models also used visual tokenizers to discrete the images and trained with causal language modeling loss. I noticed the authors cite the SEED-LLaMa in the line 102, but the claim of “In this work, we design our framework based on the autoregressive next-token prediction method for visual generation and make our VLM learn to generate visual content effectively.” does not the main difference between VILA-U and SEED-LLaMa.

2. One of the claimed contributions of this paper is about proposing the training strategy for the unified foundation vision tower. However, the training strategy seems similar to SEED [3], which also used contrastive loss between image embeddings and text embeddings. Can authors clarify the difference between the unified foundation vision tower and SEED?

3. Comparisons with other tokenization-based multimodal models [1,2] and Emu2 [4] are missing.

4. The limitation section, which is required, is missing.

[1] Zhan, Jun, et al. "Anygpt: Unified multimodal llm with discrete sequence modeling." arXiv preprint arXiv:2402.12226 (2024).

[2] Ge, Yuying, et al. "Making llama see and draw with seed tokenizer." arXiv preprint arXiv:2310.01218 (2023).

[3] Ge, Yuying, et al. "Planting a seed of vision in large language model." arXiv preprint arXiv:2307.08041 (2023).

[4] Sun, Quan, et al. "Generative multimodal models are in-context learners." Proceedings of the IEEE/CVF Conference on Computer Vision and Pattern Recognition. 2024.

**Questions:**

Please refer to the weaknesses section.

---

### Official Review · Reviewer_7Smq · 2024-11-04

**Soundness:** 4
**Presentation:** 3
**Contribution:** 3
**Rating:** 8
**Confidence:** 4

**Summary:**

- The paper presents VILA-U, a unified model for language, image and video understanding + generation
- The model is trained with an autoregressive next token prediction loss for all tasks
- The paper explores vision encoder choices to ensure understanding and generation performance

**Strengths:**

- The paper's most interesting contribution is the unified vision tower exploration to unify generation and understanding and the appropriate ways to train such an encoder
- The approach is quite straightforward and the application of RQ-VAE allows for token efficiency while preserving more information
- VILA-U is close to SOTA on visual understanding tasks (image and video) with comparable models
- The model also fares well on image generation tasks and comes close to diffusion models

**Weaknesses:**

- The method chooses RQ-VAE for efficiency, but there isn't a discussion / results around this. How would the results look if the vision tower didn't use RQ-VAE? How important is the RQ-VAE?
- The generated images are relatively low-resolution (256 or 384px), especially since the RQ-VAE allows for increased efficiency in tokens
- The paper doesn't really discuss video implementation details. Video understanding and generation have a mismatch in FPS / durations they usually support, what does VILA-U support? There isn't a discussion around this.
- The paper claims to support video generation, but there are no quantitative results around this. The two qualitative examples are also very simplistic in Figure 7.

**Questions:**

- Please share missing details as mentioned in the weaknesses
- What are the number of image and video tokens going into the LLM? How many tokens are processed by the RQ-transformer and what is its size (the RQ-VAE paper has multiple different settings)?
- It would be interesting to see if the vision tower training results hold for a general VAE setup instead of an RQ-VAE since that would make the results even more broadly applicable

---

### Meta-Review · Area_Chair_B5NY · 2024-12-20

**Metareview:**

VILA-U presents a unified foundation model that integrates video, image, and language understanding and generation within a single autoregressive next-token prediction framework. Unlike traditional visual language models that use separate modules for understanding and generating visual content, VILA-U employs a unified vision tower that aligns discrete visual tokens with textual inputs during pretraining to enhance visual perception. This innovative approach eliminates the need for additional components like diffusion models, simplifying the model architecture while maintaining high performance across tasks.

The model's effectiveness is demonstrated through comprehensive experiments across multiple benchmarks, where it achieves performance comparable to state-of-the-art specialized models in both understanding and generation tasks. Key to its success is the unified vision tower's ability to capture both semantic and appearance features through residual quantization and a combination of reconstruction and contrastive losses during training. The model also demonstrates strong in-context learning capabilities and can handle interleaved sequences of images and text effectively.

**Additional Comments On Reviewer Discussion:**

The main concerns raised by reviewers included:
- Reviewer 7Smq questioned the effectiveness of residual quantization and requested more details about video implementation;
- Reviewer ma7u asked for clarification on differences between VILA-U and other tokenization-based models like AnyGPT and SEED-LLaMa;
- Reviewer X72f and n4tc questioned the positioning of the work's novelty and whether improvements came from data engineering rather than architectural advances;
- Reviewer n4tc mainly asked about the in-context learning abilities of VILA-U;
- Reviewers also requested additional comparisons with models like OpenSora and CogVideoX for video generation tasks.

The authors adequately addressed all these concerns during the rebuttal period, ultimately leading to four positive ratings, by (1) Providing ablation studies demonstrating the significant benefits of residual quantization over standard vector quantization, (2) Clarifying that VILA-U's unified vision tower differs from previous approaches by combining both semantic understanding and generation capabilities without requiring external models, (3) Emphasizing that no special data curation was performed and improvements came from architectural innovations like the unified vision tower and residual quantization, and (4) Adding comprehensive comparisons with additional baselines including OpenSora and CogVideoX on the VBench benchmark.

---

### Decision · Program_Chairs · 2025-01-22

Accept (Poster)